# Artificial Intelligence in CT and MR Imaging for Oncological Applications

**DOI:** 10.3390/cancers15092573

**Published:** 2023-04-30

**Authors:** Ramesh Paudyal, Akash D. Shah, Oguz Akin, Richard K. G. Do, Amaresha Shridhar Konar, Vaios Hatzoglou, Usman Mahmood, Nancy Lee, Richard J. Wong, Suchandrima Banerjee, Jaemin Shin, Harini Veeraraghavan, Amita Shukla-Dave

**Affiliations:** 1Department of Medical Physics, Memorial Sloan Kettering Cancer Center, New York City, NY 10065, USA; 2Department of Radiology, Memorial Sloan Kettering Cancer Center, New York City, NY 10065, USA; 3Department of Radiation Oncology, Memorial Sloan Kettering Cancer Center, New York City, NY 10065, USA; 4Head and Neck Service, Department of Surgery, Memorial Sloan Kettering Cancer Center, New York City, NY 10065, USA; 5GE Healthcare, Menlo Park, CA 94025, USA; 6GE Healthcare, New York City, NY 10032, USA

**Keywords:** artificial intelligence, cancer, computed tomography, deep learning, diffusion-weighted magnetic resonance imaging, radiomics

## Abstract

**Simple Summary:**

The two most common cross-sectional imaging modalities, computed tomography (CT) and magnetic resonance imaging (MRI), have shown enormous utility in clinical oncology. The emergence of artificial intelligence (AI)-based tools in medical imaging has been motivated by the desire for greater efficiency and efficacy in clinical care. Although a growing number of new AI tools for narrow-specific tasks in imaging is highly encouraging, the effort to tackle the key challenges to implementation by the worldwide imaging community has yet to be appropriately addressed. In this review, we discuss a few challenges in using AI tools and offer some potential solutions with examples from lung CT and MRI of the abdomen, pelvis, and head and neck (HN) region. As we advance, AI tools may significantly enhance clinician workflows and clinical decision-making.

**Abstract:**

Cancer care increasingly relies on imaging for patient management. The two most common cross-sectional imaging modalities in oncology are computed tomography (CT) and magnetic resonance imaging (MRI), which provide high-resolution anatomic and physiological imaging. Herewith is a summary of recent applications of rapidly advancing artificial intelligence (AI) in CT and MRI oncological imaging that addresses the benefits and challenges of the resultant opportunities with examples. Major challenges remain, such as how best to integrate AI developments into clinical radiology practice, the vigorous assessment of quantitative CT and MR imaging data accuracy, and reliability for clinical utility and research integrity in oncology. Such challenges necessitate an evaluation of the robustness of imaging biomarkers to be included in AI developments, a culture of data sharing, and the cooperation of knowledgeable academics with vendor scientists and companies operating in radiology and oncology fields. Herein, we will illustrate a few challenges and solutions of these efforts using novel methods for synthesizing different contrast modality images, auto-segmentation, and image reconstruction with examples from lung CT as well as abdome, pelvis, and head and neck MRI. The imaging community must embrace the need for quantitative CT and MRI metrics beyond lesion size measurement. AI methods for the extraction and longitudinal tracking of imaging metrics from registered lesions and understanding the tumor environment will be invaluable for interpreting disease status and treatment efficacy. This is an exciting time to work together to move the imaging field forward with narrow AI-specific tasks. New AI developments using CT and MRI datasets will be used to improve the personalized management of cancer patients.

## 1. Introduction

Most common high-resolution cross-sectional anatomic imaging modalities, such as computed tomography (CT) and magnetic resonance imaging (MRI), excel at providing details regarding lesion location, size, morphology, and structural changes to adjacent tissues [1]. There is abundant literature on qualitative and quantitative CT and MRI focusing on oncological applications [2,3]. Such images capture features, e.g., tumor density, enhancement pattern, margin irregularity, and relation to neighboring structures, which are then used for tumor detection, initial cancer staging, assessment of treatment response, and clinical follow-up [4]. For example, in routine clinical trials, radiologists provide lesion size measurements using Response Evaluation Criteria in Solid Tumors (RECIST) guidelines for medical oncologists and radiation oncologists to assess treatment response [5]. Such size measurements are labor-intensive and can be replaced by new auto-segmentation tools that help to calculate tumor volume in a more accurate, reproducible, and time-efficient manner [6,7]. The primary driver behind the emergence of artificial intelligence (AI) in medical imaging has been the desire for greater efficacy and efficiency in clinical care [8,9]. The topics of data sampling and deep learning (DL) strategies, including levels of learning supervision (transfer learning, multi-task learning, domain adaptation, and federated and continuous learning systems), are well covered in previously published reviews [10,11]. The importance of proper data collection and standardization methods, the appropriate choice of the reference standard in relation to the task at hand, the identification of suitable training approaches, the correct selection of performance metrics, the requirements of an efficient user interface, clinical workflows, and timely quality assurance of AI tools cannot be emphasized enough [12,13]. The imaging community must address the challenges together and identify target areas that can benefit from AI opportunities. Present challenges include testing the accuracy and reliability of quantitative CT and MRI data before its inclusion in the AI pipeline as well as how best to integrate AI developments into clinical practice [14,15].

Here, we will illustrate a few challenges and solutions of these efforts using novel methods for synthesizing different contrast modality images, auto-segmentation and image reconstruction with examples from lung CT as well as abdomen, pelvis, and head and neck MRI. Discussion of AI developments in other imaging modalities, including X-ray, mammography, ultrasonography, and positron emission tomography (PET), is beyond the scope of this review.

### 1.1. Highlights

AI applications in CT and MRI oncological imaging may be leveraged for protocol development, imaging acquisition, reconstruction, interpretation, and clinical care.

Herein are highlighted the key points of the review:
o Deep learning methods can be used to synthesize different contrast modality images for many purposes, including training networks for multi-modality segmentation, image harmonization, and missing modality synthesis.o AI-based auto-segmentation for discerning abdominal organs is presented here. Deep learning methods can leverage different modalities with more information (e.g., higher contrast from MRI or many experts segmented labeled datasets such as from CT) to improve tumor segmentation performance in a different modality without requiring paired image sets.o Deep learning reconstruction algorithms are illustrated with examples for both CT and MRI. Such approaches improve image quality, which aids in better tumor detection, segmentation, and monitoring of response.o It is emphasized that large quantities of data are requirements for AI development, and this has created opportunities for collaboration, open team science, and knowledge sharing.

### 1.2. AI in CT and MRI for Oncological Imaging

AI tools represent a potential leap forward in oncological imaging, including harnessing machine learning and DL to improve tumor characterization, identify imaging biomarkers for histopathological, metabolic, and functional status, and tailor treatment plans [16]. AI methods have shown the potential to stratify patients based on risk factors as well as provide automated measurements of tumor volume via tumor segmentation [10,15,17]. Many studies have been published on machine learning tools for computer-aided or AI-assisted clinical tasks [8,9,11,18]. However, most of these tools are not yet ready for clinical deployment. It is of paramount importance that any AI-driven clinical tool undergo proper training and rigorous validation of its generalizability and robustness before being adopted into patient clinical care [15,19,20,21].

Highly accurate tumor segmentation would allow for reliable and reproducible longitudinal tracking of tumor size and volume across time points. Automated segmentation can be easily integrated into clinical oncological imaging workflows, overcoming the time limitations of manual size comparisons [22,23]. Although RECIST remains the standard methodology for clinical trials, it is difficult to implement in daily clinical practice [5]. Furthermore, rapid progress in computational power and new AI techniques can allow for the processing of larger data sets to reveal new imaging biomarkers that are surrogates for tumor subtypes and disease status [24]. AI models can now be constructed incorporating the full spectrum of clinical, genomic, and histopathologic data in tumor classification [25], tumor subtyping with non-invasive quantitative imaging data, and tumor histopathology. Lastly, genomics data can revolutionize cancer management by guiding treatment selection and determining prognosis [26,27].

AI efforts in CT and MRI are already well underway and have demonstrated remarkable progress in various image analysis tasks [8,9,10,11]. In cancer screening, DL techniques have shown promise in CT screening for lung cancer and colonic polyps [28], MRI screening for prostate cancer [29], discriminating glioblastoma from brain metastasis with conventional MR images [30], breast cancer risk assessment with MR images [31,32], and segmentation of CT and MR images of head and neck (HN) cancer for MR-guided radiotherapy [33,34,35]. AI models trained on large datasets can extract high-dimensional representations, which show an increase in specificity compared with lower-dimensional machine learning methods often used in computer-aided detection software for lung cancer screening [36]. The advent of precision medicine in oncology aims to tailor individual treatment plans based partly on tumor genomics and histopathology [37]. Typically, this data is obtained through invasive procedures. However, the ability to non-invasively capture such data can augment precision medicine with radiomics and therefore change clinical management. In neuro-oncology, in particular, research efforts aim to predict the presence of IDH1 mutations, 1p/19q co-deletion, and EGFR, as well as VEGF and p53 status, by identifying precise imaging biomarkers via machine learning and DL techniques [38]. Tumor subtyping may further aid the determination of cancer prognosis [39]. Attempts have been made using AI tools to predict survival outcomes in glioblastoma multiforme based on baseline brain MRI [40] as well as to predict response to chemoembolization in hepatocellular carcinoma (based on baseline liver MRI [41]. A comprehensive understanding of the invasive histopathological and molecular approaches, which provide insight into intratumor heterogeneity and the role of advanced MRI imaging in characterizing microstructures, cellularity, physiology, perfusion, and metabolism, is lacking [42,43]. Thus, developing informed, cutting-edge, robust AI tools using imaging datasets is necessary to quantify imaging biomarkers and improve patient diagnostics and outcomes.

## 2. Specific-Narrow Tasks Developed Using AI for Radiological Workflow

Figure 1 illustrates the many opportunities for specific-narrow tasks developed using AI in radiological workflow, which range from imaging protocol development and data acquisition to the interpretation of images for clinical care. AI can be helpful in patient protocol systems, starting with selecting proper imaging tests depending on the organ under study, exam scheduling, protocoling, and retrieving available prior images for comparison. All major imaging vendors incorporate AI, which shows great promise for patient positioning [44], image acquisition, and reconstruction pipelines by reducing scan time, suppressing artifacts, and improving overall image quality via optimization of the signal-to-noise ratio (SNR) [45,46]. AI-based image reconstruction methods can also help minimize the radiation dose from CT images by improving image quality [23,35]. AI tools developed for specific, narrow tasks, such as case assignment, lesion detection, and segmentation of regions of interest, are critical for oncological imaging. Reconstruction of images using DL algorithms has shown remarkable improvements in image contrast and SNR for CT [19,47,48] and MRI [49,50,51]. As mentioned above, manually segmenting longitudinal tumor volume is laborious, time-consuming, and difficult to perform accurately. Previously developed auto-segmentation methods were sensitive to changes in scanning parameters, resolution, and image quality, which limited their clinical value [52]. AI-based algorithms have been successful at tumor segmentation and have shown better accuracy and robustness to imaging acquisition differences [49,50,51]. In parallel, new AI tools have been developed for the quantification of image features from both radiomics and lesion classification [16,53,54]. AI models could help integrate multi-modality imaging data and molecular markers as available [25]. AI methods are also amenable to developing predictive and prognostic models for clinical decision-making and/or clinical trials [55]. With these developments, AI is poised to be the main driver for innovative CT and MR imaging, and it can play an important role in clinical oncology.

This is an exciting time for imaging professionals, in which radiologists and scientists will remain essential for producing the highest quality imaging data and its interpretation for clinical care. Herein, we illustrate the challenges of CT and MR image analysis using AI tools as well as offer some potential solutions originating from our experience using examples from lung CT and MRI of the abdomen, pelvis, and HN region.

## 3. Major Challenges with Solutions for Radiological Image Analysis

The major challenges in radiological image analysis are described pointwise with solutions in this section. Accordingly, we have summarized a selection of original and review articles with references, the narrow-specific AI tasks, title, objectives, advantages, recommendations, and limitations (if applicable) in Table 1. The select articles from 2018 to 2022 cover AI applications and their use in (i) medical imaging, (ii) image reconstruction and registration, (iii) lesion segmentation, detection, and characterization, and (iv) clinical applications in oncology. It was beyond the scope of this work to include the full list of articles published in this area.

### 3.1. Variability in Imaging Acquisition Pose Challenges for Large-Scale Radiomics Analysis Studies

Radiomics, or the non-invasive extraction of quantitative information from images, is well developed in oncology, with several groups demonstrating its utility for both cancer diagnosis and treatment response prediction of multiple solid cancers [62]. However, these successes have yet to be translated into routine clinical use due to the variability in MRI [63] and CT [64] images stemming from varying image acquisition protocols and multi-vendor scanners that affect radiomics features. Hence, cross-site image harmonization remains an urgent, unmet need to enable robust multi-institutional and clinical use of radiomics biomarkers.

Commonly used image harmonization methods, such as ComBat, use the statistical properties of the data distribution to reduce the variability of radiomic features by removing so-called “batch effects” by shifting [65] distributions and using the unrealistic assumption of a unimodal feature distribution. Multi-modality of feature distributions can be addressed with multiple mixture Gaussian-based ComBat [66] normalizations, but such methods still require pre-determined groupings and a fixed set of features.

Recent developments in domain adaptation using generative adversarial networks (GANs) have successfully applied image harmonization to CT and MRI images [67,68,69,70]. However, such methods have limited success due to their reliance on global image similarity losses, which can lead to the introduction of unexpected artifacts and hallucinated features as well as the potential loss of diversity in the textural content. Disentangling DL methods, which extract domain-invariant content such as tumor shape, anatomic context, and domain-specific style, are more robust to domain differences and best mitigate mode collapse issues [12,71]. However, DL methods also require the training of multiple one-to-one modality mapping methods, which increases the need for computational and memory capacity to accommodate a variety of scanner and imaging protocols.

Other prior works have used GANs for image synthesis for a variety of purposes [72,73,74,75,76,77], including generating PET images from CT using bi-directional contrastive GANs constructed to maximize the information between two networks generating CT to PET and PET to CT images, respectively. To generate missing PET images, synthesizing liver contrast to improve tumor detection by combining a GAN with a self-attention convolutional network and a region-based discriminator to improve tumor segmentation [77], multi-contrast MRI generation using CTs with the so-called MedGAN for medical imaging applications [73], as well as ensuring realistic texture preservation with texture preservation losses implemented into the GAN network training [72]. Whereas the aforementioned methods focused on preserving textural characteristics and inverse consistency to ensure synthesis, other works used attention formulations to focus the network towards regions or structures of interest. One technique, SAGAN [74], uses region masks to provide additional constraints. Another technique, PSIGAN, combines derived structure information using a jointly trained segmentation and image synthesis network for learning to segment on MRI images without labeled MRI datasets [78]. Recently, a new CVT-GAN method combined a convolutional framework with vision transformers to extract global and local self-attention methods for high-quality standard-dose PET (SPET) reconstruction using low-dose PET (LPET) images [76].

In prior work, a disentangled deep network approach was developed that employs a single universal content encoder with a single variational autoencoder to extract both image content and style for one-to-one domain adaptation [75]. Using our approach, a style code is extracted from the images and converted into latent style codes that can then be used to modulate image generation. A key difference between our variational auto-encoder approach and other prior methods is that our method learns a one-to-many modality translation using a lightweight scaling module that extracts the style code for the different modalities as a scaling function, which is then injected into a single decoder to generate the different modality images. Therefore, our approach makes use of a smaller memory footprint architecture consisting of a single domain in-variant content encoder, a lightweight style coder network, and a single decoder network. Other methods require multiple one-to-one modality synthesis networks for every single considered modality [72,73,74,75,76,77].

Extensive details of our method have been published in several outlets [59,75,79,80]. Briefly, our method includes a domain-invariant content encoder network composed of a sequence of convolutional layers and a single style coding network that extracts the latent style code for the different modalities. The style coding network is constructed using a variational autoencoder, which uses a latent Gaussian prior to span the styles of the various modalities and is constructed using 5 convolutional pooling layers, followed by a global pooling and fully connected layer. The style code is transformed into a latent style scale by a latent scale layer that is then used to modulate the features computed by the decoder network to synthesize images corresponding to different modalities. This network is jointly optimized using adversarial losses using a patchGAN discriminator, content reconstruction losses, image translation losses, and latent code regression losses as detailed in prior work [75]. In addition, a multi-tasked training strategy is used in which a two-dimensional (2D) Unet architecture is employed to learn to generate multi-organ segmentation from the synthesized image sets. The networks are optimized using the Adam method with a batch size of 1 and a learning rate of 2 × 10^−4^, with early stopping used to prevent overtraining [81].

The result of synthesized T2-weighted (T2w) MRI into T1-weighted (T1w) MRI from CT datasets available in the open-source Combined Healthy Abdominal Organ Segmentation (CHAOS) challenge dataset are shown in Figure 2. Using a published method described by Jiang and Veeraraghavan [75], the model was trained using 20 unlabeled MRIs and an entirely different set of 30 patients with expertly segmented CT images containing multiple organ segmentations. Testing was performed on another group consisting of 10 patients who had undergone MRI exams. Both sequences were acquired on a 1.5 Tesla scanner. As shown, our approach produced a realistic synthesis of such images, indicating potential use in image harmonization.

Synthesis realism was measured by computing the similarity between the features computed within the individual organs on synthesized images and those same organs in real images. Our method produced a low distance of 5.05 and 14.00 for T1w and T2w MRI. In comparison, this distance was 73.90 and 101.37 for T1w and T2w MRI using CycleGAN, which learns multiple one-to-one modality translations, and 73.39 and 77.49 using another state-of-the-art one-to-one modality translation method called StarGAN [82].

### 3.2. Volumetric Segmentation of Tumor Volumes and Longitudinal Tracking of Tumor Volume Response

Currently, radiographic response assessment during treatment and at follow-up is primarily applied using bi-dimensional RECIST metrics [5], which have many limitations and cannot quantify the underlying phenotypic heterogeneity within tumors. For practical use, automated and consistent pipelines for quantifying longitudinal tumor response dynamics are needed. Reliable segmentation is also necessary to overcome the practical limitations of radiomics analysis methods, which require volumetric tumor segmentation.

Recent works have shown the possibility of obtaining a more accurate tumor prognosis by utilizing longitudinal tumor response image features extracted from radiomics analysis [54,83,84,85,86]. Multi-tasked AI methods that combine segmentation and classification of serial images have shown the ability to predict tumor treatment response for rectal cancers better [87]. In this context, AI-enabled longitudinal image analysis is needed to both segment and characterize tumor changes at the voxel level. Containerized and operating system-independent segmentation tools, such as DeepNeuro [88] and DeepInfer [89], provide well-known AI models for specific disease sites, primarily brain and prostate cancers. Community supported resources, such as MONAI [90], have increased the ability to extract, transform, and load data for tailored DL model development, thereby lowering the barrier to DL tool assessment for the general research community.

These successes have spurred growth in offering commercial tools for normal tissue segmentations for several disease sites. However, successes in normal tissue segmentation and a few cancers, such as brain gliomas, have yet to be translated to tumors in other disease sites and imaging modalities, such as contrast-enhanced and non-contrast CTs and cone-beam CTs that are routinely used in radiotherapy. New DL methods that learn the underlying spatial anatomic context, including those that use vision transformers and self-attention methods [91,92] have improved the ability of DL to extract the segmentation of challenging tumors. Another related recent innovation is the development of distillation learning and cross-modality learning [45,93,94], in which information from different modalities, such as CT or MRI, is used to inform and improve the extraction of relevant features that better signal the contrast between tumor and background. In addition to improving segmentation in imaging modalities with low soft-tissue contrast, such as CT and cone-beam CT, using the information learned from higher contrast modalities (e.g., MRI) can also benefit learning in new modalities for disease sites (such as MRI for the lung), in which expertly segmented datasets are limited [95].

Figure 3 shows the results with example segmentations produced by a cross-modality educed distillation learning method (CMEDL) [79], which combines learning from unpaired or unrelated sets of T2w turbo spin echo MRI and CT as well as cone beam computed tomography (CBCT) images for the segmentation of lung tumors. Segmentation on T2wMRI produced via unpaired distillation learning, in which many CT datasets (*n* = 300) relative to MRI datasets (*n* = 80) were available, demonstrates the additional use case of unpaired distillation learning for data augmentation. The results shown in Figure 3A–C are produced by three different models that were trained using the CMEDL approach. Extensive details of the CMEDL method are in the prior published methods for CT lung tumor [96], MRI lung tumor segmentation [79], and CBCT-based lung tumor segmentation [59]. Concisely, the CMEDL architecture makes use of two parallel segmentation subnetworks for a so-called tracker network (using MRI [Figure 3A,B], CT [Figure 3C]), and a student network (using CT [Figure 3A], CBCT [Figure 3B], and T2w MRI [Figure 3C]). Any segmentation architecture can be used, as shown using the popular Unet as well as a dense network called a multiple resolution residual network [97]. The teacher network forces the student network to extract features that better signal the contrast between foreground and background by applying feature distillation losses that match the high-level features computed from corresponding synthesized teacher modality (e.g., MRI) and student modality (e.g., CT) images.

The network itself is trained with unpaired images, in which corresponding sets of multiple-modality scans are not required for training. To accomplish training with unpaired modalities, a cross-modality synthesis network created using a GAN is applied. The GAN consists of a generator created using a 3DUnet that computes dense pixel regression by using *tanh* activation, and a PatchGAN discriminator network to distinguish the synthesized from the real images was used in training. The details of the number of images used in training, training losses, training epochs, etc. are in published methods [79]. The teacher network is initialized with example real images and corresponding segmentations to learn to extract the appropriate set of relevant features. The same network is then jointly optimized with the student network to further refine the extracted features using synthesized images produced from the images input to the student network using the GAN-based image-to-image translation network. The teacher and student networks are jointly optimized during training to make use of multi-task optimization. The GAN network for synthesizing the cross-modality images is cooperatively optimized such that this network’s parameters are updated only in iterations when the segmentation network’s parameters are frozen, and vice versa, to ensure stable training convergence.

The results of segmenting the tumor on CT images using a Unet network on a sample test case and optimized via the CMEDL approach with CTs (*n* = 377) and MRIs (*n* = 82) from external and internal institution datasets, respectively, are shown in Figure 3A. The results of segmenting an external institution CBCT image using a Unet network optimized with the CMEDL approach optimized with unpaired CBCTs (*n* = 216) and 82 MRIs from different sets of patients are shown in Figure 3B. Figure 3C shows a sample test-set MRI segmentation produced by training a Unet using the CMEDL approach. Separate models were constructed for the three results and optimized with different datasets. All networks were optimized with the Adam optimizer, with an initial learning rate of 2 × 10^−4^, batch size of 2, and early stopping was used to prevent overfitting of the networks. As shown in Figure 3, the algorithm generated segmentations closely approximates the expert delineation for the representative test cases.

Although the aforementioned method focuses on the segmentation of the gross tumor volume (GTV), it is also important to consider the tumor margin needed for effective treatment when using the AI-defined tumors for treatment planning and delivery [59,78,79,97,98]. For instance, in the context of thermal ablation, prior work by Singh et al. [99] showed that incorporating blood perfusion information from dynamic contrast MRI using commercial software tools could be utilized to better define the margins of breast tumors for thermal ablation. In the context of radiation therapy, the segmented GTV is often expanded to produce a clinical target volume (CTV) to incorporate the microscopic spread by using treatment planning software to generate automatic expansion with fixed criteria for different disease sites while aiming to limit radiation exposure to the adjacent healthy tissues. However, this approach does not always account for microscopic disease, and hence, it is resolved using a clinician’s manual delineation that leads to inter-rater variability [100]. Cardenas et al. [101] addressed this issue of clinical variability by using a stacked autoencoder deep network formulation to automatically learn the CTV definition for head and neck cancers while accounting for adjacent healthy tissues both for lymph nodes and high-risk CTV. A different prior work by Xie et al. [102] addressed the issue of lung cancer CTV definition by accounting for respiration and GTV contained within the CTV by constructing a customized loss function within a 3DUnet approach.

### 3.3. Optimization of Dose and Image Quality Improvement in CT Scans

CT is an essential component of modern healthcare [103,104]. With technical improvements, such as iterative reconstruction (IR) [105], dual-energy CT [106], ultra-high resolution CT [107], and the latest innovation of photon counting CT [106], the spectrum of potential clinical applications has dramatically increased [103]. Nevertheless, there is still much to be done to reduce radiation exposure while suppressing noise and preserving or improving spatial and contrast resolution [103,105,108,109]. Although current model-based IR algorithms and their variants compensate for the increased noise caused by reduced radiation doses, the shifted image texture with IR relative to conventional filtered back projection is subjectively inferior and less preferred by radiologists [108,109,110].

To address this challenge and democratize technology, researchers have looked to AI- or DL-based image reconstruction solutions to improve imaging capabilities while reducing radiation doses [111]. DL-based CT reconstruction (DLR) has emerged as a promising alternative to conventional CT reconstruction methods [109,112]. Several literature reports demonstrate DLR to be superior to IR at noise suppression and artifact reduction [113,114,115]. Therefore, radiologists subjectively prefer DLR for several diagnostic tasks [113,116]. One commercially available DL-based solution, TrueFidelity (General Electric Healthcare [GEHC], Madison, WI, USA), trains a deep convolutional neural network (CNN) to map low-dose CT images to a higher quality and high-dose version of the same data [109,115]. TrueFidelity differentiates and suppresses noise while reconstructing CT images with characteristics resembling the higher-quality scans from the training set [109]. A recent clinical investigation reported improvements in radiologists’ subjective image quality scores as well as gains in contrast-to-noise ratio and noise reduction while reducing radiation dose by more than 50% for the detection of liver lesions >0.5 cm from portal venous abdominal CT exams [117]. DLR methods are expected to be the future of CT image reconstruction [58,118]. With improved algorithms, computational power, and more data, DL-based image reconstruction will continue outperforming model-based IR and its variants at generating low-noise images without sufficient image quality across diagnostic tasks for human viewers [58,118].

### 3.4. Optimization of Image Quality in MRI Scans

Conventional MR data acquisition methods provide excellent soft-tissue contrasts in images and are routinely used for oncological diagnostic workups. SNR and spatial resolution constraints, motion artifacts, and longer scan times can, at times, be limiting factors in MRI, depending on the organ of interest [17,49]. For cancer patients unable to stay in the MRI scanners for a half hour or longer, there is an urgent need for rapid and robust MR imaging acquisition that improves patient comfort and throughput. For example, GEHC, a major MRI vendor, recently introduced the novel DL-based MR reconstruction (Recon), AIR™ Recon DL method, which is revolutionizing MR image reconstruction for anatomical T1w- and T2w imaging by improving the image quality with high SNR, sharpness, and reduced scan time.

The AIR™ Recon DL reconstruction process converts raw k-space data into high-quality images as its output [49,119]. This new approach will generate images free of ringing artifacts and reduced noise, leading to increased diagnostic accuracy compared with conventional methods. The AIR™ Recon DL pipeline does not require resolution-degrading filters, which are commonly embedded in the traditional reconstruction pipeline. Instead, it utilizes a deep CNN that works on raw, complex-valued imaging data to produce a clear output image. The CNN has been specifically designed to allow for a user-controlled reduction in image noise, reduction of truncation artifacts, and enhancement of edge sharpness. There is also a window of opportunity for AI to both improve image quality and quantify imaging biomarkers derived from quantitative techniques, such as diffusion-weighted (DW)-MRI that measures the random Brownian motion of water molecules in tissue [120].

Recent literature has shown promise for DW-MRI powered with DL Recon in diagnostic applications for brain tumors [121], liver cancer [122], and prostate cancer [123], reporting higher SNR and image quality. We are working with GEHC scientists to apply this technology to different body organs at our center, thereby DL Recon improving diagnostic images and the robustness of imaging biomarkers. This new DW-MRI protocol will allow for modification of the MRI acquisition parameters, including b-values and the number of excitations. Figure 4 demonstrates preliminary experience with this method, and the images were acquired from patients with papillary thyroid cancer and lymphoma. A whole-body DW-MRI was performed on the lymphoma patient to detect the existence of disease spread to other vital organs.

### 3.5. Bias in AI Models

Although the growing number of new AI models for narrow-specific tasks in CT and MRI is highly encouraging, the effort to tackle key challenges to implementation by the worldwide imaging community has yet to be addressed. AI-based system pipelines consist of data sampling and DL strategies, including various levels of learning supervision, before drawing conclusions from the learned model [9,10,11,15,16]. Therefore, uncertainty and bias are important considerations when working with AI tools [12]. Uncertainty is the degree of variability in the model’s predictions, although the bias is a systematic error in the model. However, inherent uncertainties and biases are associated with each step that arises in data collection, noise in the data, and modeling approaches with AI tools [124]. Reproducibility assesses measurement uncertainty, which in measurement typically arises from multiple sources. It is critical that the results of AI systems are both reproducible and reliable to enable the development of personalized cancer care strategies [21,91,125].

The AI tools developed so far have shown pivotal results in providing better accuracy for prognosis, diagnosis, and assessment of treatment response using tumor characteristics obtained from radiologic images. However, these studies do not explicitly account for bias in their AI training sets [12]. Bias in AI studies remains a major challenge that must be addressed by proper data collection practices. Suboptimal data collection can introduce bias and lead to a misleading perception of model performance, especially in subpopulations that may not be appropriately represented in a study’s dataset. The data collection process must be described in detail to demonstrate scientific rigor, which requires transparent inclusion and exclusion criteria as well as the target cancer patient demographics. Unequal demographics of cancer patients and disparate access to the healthcare system due to economic inequalities impede the study of certain cancers in underrepresented populations [114]. Variability in the manifestation of cancers across subgroups can act as confounders. Access to large, high-quality datasets across low-income countries is often understudied due to a lack of research funding. Moreover, pediatric patients and young children are not smaller versions of adults and should not be studied as such. Their organ size, shape, and appearance on CT or MRI exams differ considerably from those of adult patients. An AI system that may appear functional for adult patients should not be assumed to work for pediatric patients. While accounting for potential biases, investigators may unintentionally limit their data search to their centers or within collaborating groups. One solution to this limiting factor would be to rebalance datasets by including more representative data from underrepresented communities before training AI models.

Another potential solution is to train AI systems using raw or unprocessed CT scan data. Most CT scans that train current AI systems are processed for the human visual system. As a result, the steps to generate a human-interpretable image may lead to a loss of potentially relevant information because raw data is downsampled and compressed [126]. Moreover, each vendor implements proprietary solutions to enhance the quality of their scans so that they are more appealing than their competition. These processing steps inject unique patterns unrelated to the target signal that the AI systems could spuriously use to correlate with class labels. The issues stemming from post-processed data training could be overcome by developing end-to-end AI systems with raw CT data.

## 4. Discussion

Cross-sectional CT and MRI are an integral part of the diagnostic workup. Applications of novel narrow-specific AI tasks in these imaging techniques have shown promise for data acquisition, image segmentation and registration, and assessment of tumor responses to therapy in brain tumors [30], breast [32], head and neck [33,35], liver, lung, and abdominal cancers [29,61,127]. For example, the DL method has exhibited as an effective and clinically applicable tool for the segmentation of the head and neck anatomy for radiotherapy [34]. Despite exciting advancements in the AI field, challenges to the translation of these AI-based tools into radiology practice still exist. In reviewing these challenges and potential solutions, we recommend certain strategies for the CT and MRI fields in the era of AI, including collaboration between radiologists, treating physicians, and imaging scientists. The awareness of the general accuracy of the AI model and the degree of confidence in each prediction are needed and should be well documented. Oncology professionals must communicate their imaging needs for patient management to radiologists, thus motivating research and obtaining funding to perform the necessary pilot studies. Radiology must embrace the need for quantitative CT and MRI metrics beyond lesion size measurements. Our recommendations for the application of AI in CT and MRI may apply to additional imaging modalities, such as X-ray, mammography, ultrasonography, and PET. The extraction of imaging metrics using AI should be an integral part of radiology and/or oncology workflows without impeding productivity and may be incorporated into fully automated workflow systems in the future. The longitudinal tracking and extraction of imaging metrics from registered lesions and the tumor environment using AI methods will be both efficient and productive tools for interpreting clinical follow-up. Finally, analysis of big imaging data with the representation of cancer patients from all types of demographics as well as additional sources of data, such as genomics from clinical trial analysis, is expected to create a data-driven taxonomy of cancer, which will then serve to optimize treatment decisions and improve cancer prognosis. This is the best time to work together to move the imaging field forward with narrow-specific AI tasks.

## 5. Future Directions

One of the goals of AI tool development is to introduce automated methods ethically and safely into radiology practices. Since the inception of AI, experts have predicted the potential of highly tailored AI technologies for clinical oncological applications. The benefits of AI in cancer care go beyond the optimization of established treatment strategies, but we must ensure rigorous multi-disciplinary testing of these AI models before their adoption into clinical radiology workflows. However, regulatory oversight is necessary to address quality control issues and avoid algorithmic biases.

## 6. Conclusions

In this review, a few challenges and opportunities for AI application to oncological imaging were summarized using novel methods for synthesizing different contrast modality images, auto-segmentation, and image reconstruction with examples from lung CT and abdomen, pelvis, and head and neck MRI.

The major highlights of this review were centered on the application of AI methods, which can be used for the following narrow-specific tasks: (i) to synthesize different contrast modality images for a variety of purposes, including training networks for multi-modality segmentation, image harmonization, and missing modality synthesis, (ii) auto-segmentation for discerning abdominal organs is presented here, (iii) to improve CT and MR image quality, which will aid in better tumor detection, segmentation, and monitoring of response, and (iv) has created opportunities for collaboration, open team science, and knowledge sharing.

In the era of precision medicine, there is a growing interest in improving clinical decision-making as well as time to share knowledge and work together. AI tools are being developed for narrowly-specific tasks for oncological imaging needs and may contribute significantly towards enhancing clinician workflows and clinical decision-making.

## Figures and Tables

**Figure 1 cancers-15-02573-f001:**
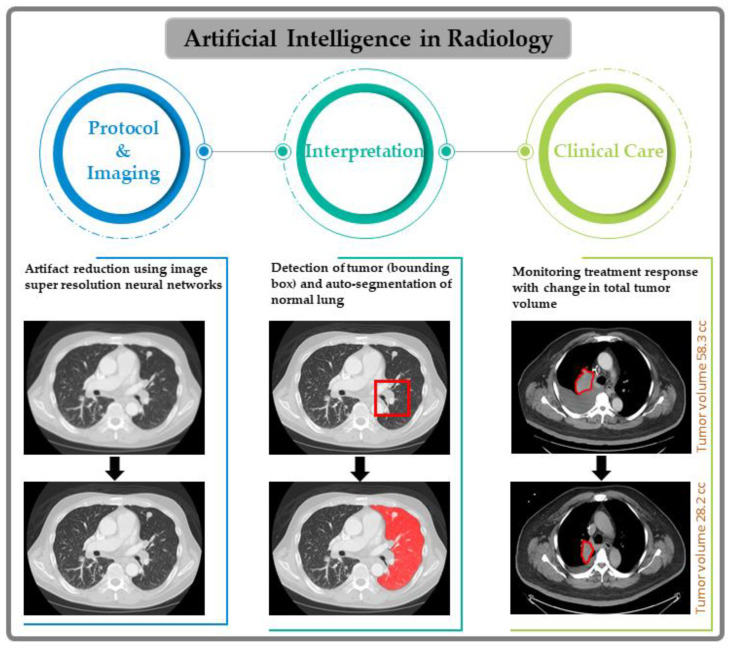
Artificial Intelligence in the clinical radiology workflow with examples from lung computed tomography.

**Figure 2 cancers-15-02573-f002:**
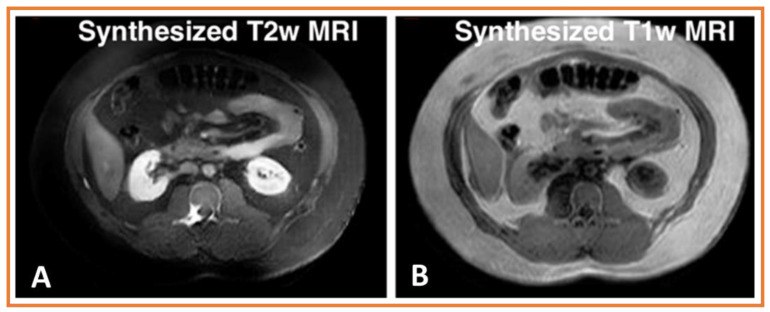
(**A**,**B**) Synthesis of T2-weighted (T2w) and T1-weighted (T1w) magnetic resonance imaging (MRI) images from computed tomography image volume available in the open-source combined healthy abdominal organ segmentation (CHAOS) challenge dataset.

**Figure 3 cancers-15-02573-f003:**
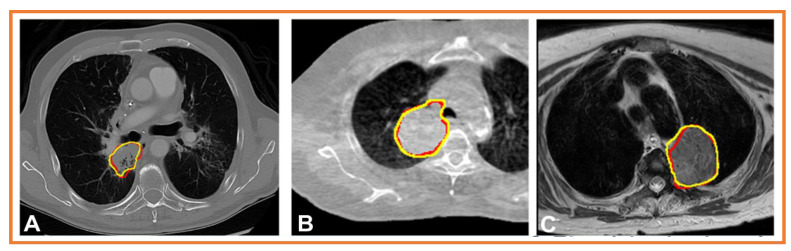
Segmentations produced by cross-modality distillation learning applied to representative cases consisting of (**A**) computed tomography (CT) image; (**B**) cone-beam CT image; and (**C**) T2-weighted magnetic resonance imaging image. Algorithm segmentations are shown in red, and the expert delineations are in yellow.

**Figure 4 cancers-15-02573-f004:**
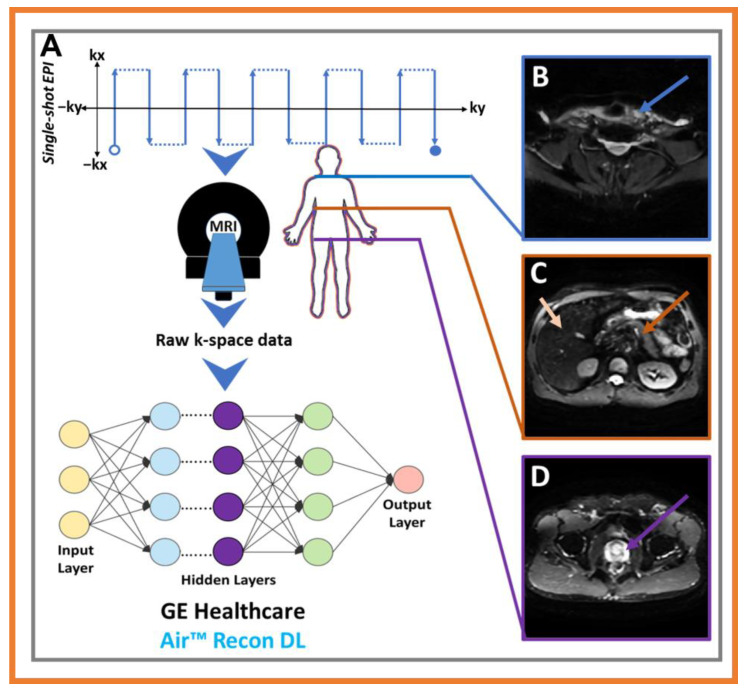
(**A**) exhibits the line diagram of the diffusion-weighted magnetic resonance image (DW-MRI) powered with deep learning recon image acquisition scheme. (**B**) DW-MR image (b = 0 s/mm^2^) acquired from a 39-year-old female patient with papillary thyroid cancer. The blue arrow points to the thyroid gland. (**C**,**D**) whole body DW-MR image (b = 0 s/mm^2^) acquired from a 61-year-old male patient with lymphoma, showing representative diffusion images from the abdomen, pelvis, including liver (light orange arrow), pancreas (dark orange arrow), and the prostate (purple arrow).

**Table 1 cancers-15-02573-t001:** Summary of Select Artificial Intelligence Literature on CT and MRI for Oncology.

Study	Narrow-Specific Tasks	Design: Title	Objective	Advantages/Recommendations	Limitations
Hosny, A. et al. [8]	Medical Imaging (MI)	Review: Artificial Intelligence (AI) in radiology	To establish a general understanding of AI methods, particularly those pertaining to image-based tasks. The AI methods could impact multiple facets of radiology, with a general focus on applications in oncology, and demonstrate how these methods are advancing the field.	There is a need to understand that AI is unlike human intelligence in many ways. Excelling in one task does not necessarily imply excellence in others. The roles of radiologists will expand as they have access to better tools. The data to train AI on a massive scale will enable a robust AI that is generalizable across different patient demographics, geographic regions, diseases, and standards of care.	Not Applicable (NA)
Koh, D.M. et al. [15]	MI	Review: Artificial Intelligence and machine learning in cancer imaging	To foster interdisciplinary communication because many technological solutions are being developed in isolation and may struggle to achieve routine clinical use. Hence, it is important to work together, including with commercial partners (as appropriate) to drive innovations and developments.	There is a need for systematic evaluation of new software, which often undergoes only limited testing prior to release.	NA
Razek, A.A.K.A. et al. [56]	MI	Review: Artificial Intelligence and deep learning of head and neck cancer	To summarize the clinical applications of AI in head and neck cancer, including differentiation, grading, staging, prognosis, genetic profile, and monitoring after treatment.	AI studies are required to establish a powerful methodology and coupling of genetic and radiologic profiles to be validated in clinical use.	NA
McCollough, C.H. et al. [57]	MI	Review: Use of Artificial Intelligence in computed tomography dose optimization	To illustrate the promise of AI in the processes involved in a CT examination, from setting up the patient on the scanner table to the reconstruction of final images.	AI could be a part of CT imaging in the future, and both manufacturers and users must proceed cautiously because it is not yet clear how these AI algorithms can be evaluated in the clinical setting.	NA
Lin, D.J. et al. [45]	Image reconstruction and registration (IRR)	Review: Artificial Intelligence for MR Image Reconstruction: An Overview for Clinicians	To cover how deep learning algorithms transform raw k-space data into image data and examine accelerated imaging and artifact suppression.	Future research needs continued sharing of image and raw k space datasets to expand access and allow for model comparisons, defining the best clinically relevant loss functions and/or quality metrics by which to judge a model’s performance, examining perturbations in model performance relating to acquisition parameters, and validating high-performing models in new scenarios to determine generalizability.	NA
McLeavy, C.M. et al. [58]	IRR	Review: The future of CT: deep learning reconstruction	To emphasize the advantages of deep learning reconstruction (DLR) over other reconstruction methods regarding dose reduction, image quality, and tailoring protocols to specific clinical situations.	DLR is the future of CT technology and should be considered when procuring new CT scanners.	NA
Jiang J. et al. [59]	Lesion segmentation, detection, and characterization (LSDC)	Original Research: Cross-modality (CT-MRI) prior augmented deep learning for robust lung tumor segmentation from small MR datasets	To develop a cross-modality (MR-CT) deep learning segmentation approach that augments training data using pseudo-MR images produced by transforming expert-segmented CT images.	The advantage of this model is that it is learned as a deep generative adversarial network and transforms expert segmented CT into pseudo-MR images with expert segmentations.	A minor limitation is the number of test datasets, particularly for longitudinal analysis, due to the lack of additional recruitment of patients.
Venkadesh, K.V. et al. [60]	LSDC	Original Research: Deep Learning for Malignancy Risk Estimation of Pulmonary Nodules Detected at Low-Dose Screening CT	To develop and validate a deep learning (DL) algorithm for malignancy risk estimation of pulmonary nodules detected at screening CT.	The DL algorithm has the potential to provide reliable and reproducible malignancy risk scores for clinicians from low-dose screening CT, leading to better management in lung cancer.	A minor limitation, the group did not assess how the algorithm would affect the radiologists’ assessment.
Bi, W.L. et al. [10]	Clinical Applications in Oncology (CAO)	Review: Artificial Intelligence in cancer imaging: Clinical challenges and applications	Highlights AI applied to medical imaging of lung, brain, breast, and prostate cancer and illustrates how clinical problems are being addressed using imaging/radiomic feature types.	AI applications in oncological imaging need to be vigorously validated for reproducibility and generalizability.	NA
Huang, S. et al. [20]	CAO	Review: Artificial Intelligence in cancer diagnosis and prognosis: Opportunities and challenges	Highlights how AI assists in cancer diagnosis and prognosis, specifically about its unprecedented accuracy, which is even higher than that of general statistical applications in oncology.	The use of AI-based applications in clinical cancer research represents a paradigm shift in cancer treatment, leading to a dramatic improvement in patient survival due to enhanced prediction rates.	NA
Diamant, A. et al. [33]	CAO	Original research: Deep learning in head & neck cancer outcome prediction	To apply convolutional neural network (CNN) to predict treatment outcomes of patients with head & neck cancer using pretreatment CT images.	The work identifies traditional radiomic features derived from CT images that can be visualized and used to perform accurate outcome prediction in head & neck cancers. However, future work could be done to further investigate the difference between the two representations.	There is no major limitation mentioned by the authors. However, they do mention that the framework used here considers the central slice, and the results could have been further improved by incorporating the entire tumor.
Liu, K.L. et al. [61]	CAO	Original research: Deep learning to distinguish pancreatic cancer tissue from noncancerous pancreatic tissue: a retrospective study with cross-racial external validation	To investigate whether CNNs can distinguish individuals with and without pancreatic cancer on CT, compared with radiologist interpretation.	CNNs can accurately distinguish pancreatic cancer on CT, with acceptable generalizability to images of patients from various races and ethnicities. Additionally, CNNs can supplement radiologist interpretation.	A minor limitation is the modest sample size.

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
