# Peer review of "Artificial Intelligence in CT and MR Imaging for Oncological Applications"

_cancers, 2023, doi:10.3390/cancers15092573_

Round 1

Reviewer 1 Report

This paper presents a study on the "Artificial Intelligence in CT and MR Imaging for Oncological Applications". This is an interesting manuscript for this journal but I suggest a major revision. Here are some bugs in this article to help the authors to profit from this article, but if the authors can't do these comments (point by point) the article will be rejected.

================================

1) General comments:

1a) Some grammatical error sees in the article. Please take time to improve the language.

================================

2) Keywords and Highlights:

2a) The authors must edit old keywords in the article.

2b) The authors must add Highlights in the article.

================================

3) Abstract:

3a) The abstract doesn’t have novelty in it. The authors should rewrite the abstract with main novelty in it.

3b) What is the main purpose of the article? The authors should focus on novelty on this section. Please highlight it.

================================

4) Introduction and Literature Review:

4a) The introduction is very short. the authors should extend the abstract's length.

4b) The authors should merge Literature Review with Introduction.

4b) The authors must proper one table and compare their articles to literature review in one table (Advantages and Disadvantages) (major comment). Not only list the author's work. They should work

================================

5) Methodology:

5a) I strongly suggest authors add the configuration methods for this method. How they config this method? (major comment).

5b) I strongly suggest authors update and modify the workflow diagram in this section (major comment).

5a) I strongly suggest authors add the control parameters for AI applcations in this section. (major comment).

================================

6) Results and Discussions:

6a) It has some figures, but technical description for figures is not enough. The authors must describe as well for every figure (major comment).

6b) this section need to update and need more detail.

6c) The authors should add more detail and use more Graphs in this sections.

================================

7) Conclusion and recommendations:

7a) Conclusion lack of novelty. Please rewrite your conclusion and add some highlight and novelty in it (major comment).

7b) Conclusion is so short the authors should extend the material.

================================

8) Graphical abstract:

8a) I think Graphical abstract can help researchers. I suggest authors add one Graphical abstract in next version

================================

9) References:

9a) References should be updates (2021-2022)

Author Response

Reply to Reviewer #1 (R1) Comments: Manuscript ID: Cancers-222018, Title: Artificial Intelligence in CT and MR Imaging for Oncological Applications

This paper presents a study on the "Artificial Intelligence in CT and MR Imaging for Oncological Applications." This is an interesting manuscript for this journal, but I suggest a major revision.

Reply to comments: We greatly appreciate your thorough reading of our invited review and providing valuable comments to improve the quality of the manuscript. This manuscript is for a special issue of Cancers: Methods and Technologies Development. We have carefully considered the comments and thoroughly revised the manuscript to address them. We have incorporated the comments and suggestions with track changes (reviewers' comments are in italics) in the revised manuscript. Below is a point-by-point response to the comments and suggestions.

R1.1) General comments:

R1.1a) Some grammatical error sees in the article. Please take time to improve the language. 

Reply to R1.1a). Thank you for pointing this out;  an additional editor has thoroughly edited the manuscript.

R1.2) Keywords and Highlights:

R1.2a) The authors must edit old keywords in the article.

Reply to R1. 2a). Thank you, the keywords have been edited accordingly.

R1.2b) The authors must add Highlights in the article.

Reply to R1.2b). Thank you for your valuable suggestion. We have added the "Highlights section" accordingly and pasted it below.

"Highlights:

AI applications in CT and MRI oncological imaging may be leveraged for protocol development, imaging acquisition, reconstruction, interpretation, and clinical care. Herein are highlighted the key points of the review:

  • Deep learning methods can be used to synthesize different contrast modality images for many purposes, including training networks for multi-modality segmentation, image harmonization, and missing modality synthesis.
  • AI-based autosegmentation for discerning abdominal organs is presented here. Deep learning methods can leverage different modalities with more information (e.g., higher contrast from MRI, or many expert segmented labeled datasets, such as from CT) to improve tumor segmentation performance in a different modality without requiring paired image sets.
  • Deep learning reconstruction algorithms are illustrated with examples for both CT and MRI. Such approaches improve image quality, which aids in better tumor detection, segmentation and monitoring of response.
  • It is emphasized that large quantities of data are requirements for AI development, and this has created opportunities for collaboration, open team science, and knowledge sharing."

R1.3) Abstract:

R1.3a) The abstract doesn't have novelty in it. The authors should rewrite the abstract with main novelty in it.

Reply to R1.3a). Thank you for pointing this out; the abstract has been revised accordingly, and the relevant text has been added in the revised manuscript emphasizing the following points: "This review manuscript highlights the application of AI, Deep learning which leverages higher image quality and improves tumor segmentation in different modalities (CT and MRI). Relevant examples have addressed these points with our published and unpublished in the application of deep learning to synthesize different contrast modality images for a variety of purposes, including training networks for multi-modality segmentation, image harmonization, and missing modality synthesis".

R1. 3b) What is the main purpose of the article? The authors should focus on novelty on this section. Please highlight it.

Reply to R1.3b). Thank you for your comment; the main purpose has been added to the revised manuscript. We have added a section to address the comment regarding highlights; please see the reply to R1.2b  for the "Highlight section."

R1.4) Introduction and Literature Review:

R1.4a) The introduction is very short. The authors should extend the abstract's length.

Reply to R1. 4a).   Thank you for your suggestions; to improve the clarity of the introduction, we have added subsection numbering: 1.0-1.2 in the revised manuscript for sections 1, 2, and 3 in the original manuscript.

R1.4b) The authors should merge Literature Review with Introduction.

Reply to R1.4b). Thank you for this helpful comment which has resulted in the addition of Table 1. Please also see the reply to R1.4a

R1.4c) The authors must proper one table and compare their articles to literature review in one table (Advantages and Disadvantages) (major comment). Not only list the author's work. They should work

Reply to R1.4c). Thank you for your valuable suggestion. We have included Table 1 with relevant key reviews and original articles, which are pasted below:

"Table 1 summarizes the original and reviews select AI papers referenced in this article highlighting AI's role and use in cross-sectional medical imaging and their use in (i) medical imaging, (ii) image reconstruction and registration, (iii) lesion segmentation, detection, and characterization, and (iv) clinical applications in oncology."

Table 1: Summary of select Artificial Intelligence literature on CT and MRI for Oncology

Study

Narrow-Specific Tasks

Design: Title

Objective

Advantages/Recommendations

Limitations

Hosny, A. et al. [1]

Medical Imaging (MI)

Review: Artificial Intelligence (AI) in radiology

To establish a general understanding of AI methods, particularly those pertaining to image-based tasks. The AI methods could impact multiple facets of radiology, with a general focus on applications in oncology, and demonstrate ways in which these methods are advancing the field.

There is a need to understand that AI is unlike human intelligence in many ways. Excelling in one task does not necessarily imply excellence in others. The roles of radiologists will expand as they have access to better tools. The data to train AI on a massive scale will enable a robust AI that is generalizable across different patient demographics, geographic regions, diseases, and standards of care.

Not Applicable (NA)

Koh, D.M. et al [2]

MI

Review:

Artificial Intelligence and machine learning

in cancer imaging

To foster interdisciplinary communication because many technological solutions are being developed in isolation and may struggle to achieve routine clinical use. Hence, it is important to work together, including with commercial partners as appropriate, to drive innovations and developments.

There is a need for systematic evaluation of new software, which often undergoes only limited testing prior to release

NA

Razek, A.A.K.A. et al [3]

MI

Review: Artificial intelligence and deep learning of head and neck cancer

To summarize the clinical applications of AI in head and neck cancer, including differentiation, grading, staging, prognosis,

genetic profile, and monitoring after treatment.

AI studies are required to establish a powerful methodology and coupling of genetic and radiologic profiles to be validated in clinical use.

NA

McCollough, C.H. et al. [4]

MI

Review: Use of Artificial Intelligence in computed tomography dose optimization

To illustrate the promise of AI in the processes involved in a CT examination, from setting up the

patient on the scanner table for the reconstruction of final images.

AI could be a part of CT imaging in the future, and both manufacturers and

users must proceed cautiously because it is not yet clear how these AI algorithms can be evaluated in the clinical

setting.

NA

Lin, D.J. et al. [5]

Image reconstruction and registration (IRR)

Review: Artificial Intelligence for MR Image Reconstruction: An Overview for Clinicians

To cover how deep learning algorithms transform raw k-space data into image data and examine accelerated imaging and artifact suppression.

Future research needs continued sharing of image and raw k space datasets to expand access and allow for model comparisons, defining the best clinically relevant loss functions and/or quality metrics by which to judge a model's performance, examining perturbations in model performance relating to acquisition parameters; and validating high-performing models in new scenarios to determine generalizability.

NA

McLeavy, C.M. et al. [6]

IRR

Review: The future of CT: deep learning reconstruction

To emphasize the advantages of deep learning reconstruction (DLR) over other reconstruction methods regarding dose reduction, image quality, and tailoring protocols to specific clinical situations.

DLR is the future of CT technology and should be considered when procuring new CT scanners.

NA

Jiang J. et al. [7]

Lesion segmentation, detection, and characterization (LSDC)

Original Research: Cross-modality (CT-MRI) prior augmented deep learning for robust lung tumor segmentation from small MR datasets

To develop a cross-modality (MR-CT) deep learning segmentation approach that augments training data using pseudo-MR images produced by transforming expert-segmented CT images.

The advantage of this model is that it is learned as a deep generative adversarial network and transforms expert-segmented CT into pseudo-MR images with expert segmentations.

A minor limitation is the number of test datasets, particularly for longitudinal analysis, due to the lack of additional recruitment of patients.

Venkadesh, K.V. et al [8]

LSDC

Original Research: Deep Learning for Malignancy Risk Estimation of Pulmonary Nodules Detected at Low-Dose Screening CT

To develop and validate a deep learning (DL) algorithm for malignancy risk estimation of pulmonary nodules detected at screening CT.

The DL algorithm has the potential to provide reliable and reproducible malignancy risk scores for clinicians from low-dose screening CT, leading to better management in lung cancer.

 A minor limitation, the group did not assess how the algorithm would affect the radiologists' assessment.

Bi, W.L. et al.[9]

Clinical Applications in Oncology (CAO)

Review: Artificial intelligence in cancer imaging: Clinical challenges and applications

Highlights AI applied to medical imaging of lung, brain, breast, and prostate cancer and illustrate how clinical problems are being addressed using imaging/radiomic feature types.

AI applications in oncological imaging need to be vigorously validated for reproducibility and generalizability.

NA

Huang, S. et al. [10]

CAO

Review: Artificial intelligence in cancer diagnosis and prognosis: Opportunities and

challenges

Highlights how AI assists in cancer diagnosis and prognosis, specifically

about its unprecedented accuracy, which is even higher than that of general statistical applications in

oncology.

The use of AI-based applications in clinical cancer research represents a paradigm shift in cancer treatment, leading to a dramatic improvement in patient survival due to enhanced prediction rates.

NA

Diamant, A. et al. [11]

CAO

Original research: Deep learning in head & neck cancer outcome prediction

 To apply convolutional neural network (CNN) to predict treatment outcomes of patients with head & neck cancer using pretreatment CT images.

 The work identifies traditional radiomic features derived from CT images that can be visualized and used to perform accurate outcome

prediction in head & neck cancers. However, future work could be done to further investigate the difference between the two representations.

There is no major limitation mentioned by the authors. However, they do mention that the framework used here considers the central slice, and the results could have been further improved by incorporating the entire tumor.

Liu, K.L. et al. [12]

CAO

Original research: Deep learning to distinguish pancreatic cancer tissue from noncancerous pancreatic tissue: a retrospective study with cross-racial external validation

To investigate whether CNNs can distinguish individuals with and without pancreatic cancer on CT, compared with radiologist interpretation.

CNNs can accurately distinguish pancreatic cancer on CT, with acceptable generalizability to images of patients from various races and ethnicities. Additionally, CNNs can supplement radiologist interpretation.

A minor limitation is the modest sample size.

R1.5) Methodology:

R1.5a) I strongly suggest authors add the configuration methods for this method. How they config this method? (Major comment).

Reply to R1.5a). Thank you for this helpful comment and the opportunity to clarify that this is not a methods paper but an invited review. We have added relevant details describing the networks in the revised manuscript and the appropriate references for completeness. The method detailed in the review is based on published work stemming from 2 journal articles [7,13] and two 8-page conference publications [14-19]. The additions are pasted below.

" Briefly, our method includes a domain invariant content encoder network composed of a sequence of convolutional layers and a single style coding network that extracts the latent style code for the different modalities. The style coding network is constructed using a variational autoencoder, which uses a latent Gaussian prior to span the styles of the various modalities and is constructed using 5 convolutional, pooling layers, followed by a global pooling and fully connected layer. The style code is transformed into a latent style scale by a latent scale layer that is then used to modulate the features computed by the decoder network to synthesize images corresponding to different modalities. This network is jointly optimized using adversarial losses using a patchGAN discriminator, content reconstruction loss, image translation losses, and latent code regression losses, as detailed in prior work [19]. In addition, a multi-tasked training strategy is used wherein which a  two-dimensional 2D Unet architecture is employed to learn to generate multi-organ segmentation from the synthesized image sets. The networks are optimized using the Adam method, a batch size of 1, and learning rate of 2e-4, with early stopping used to prevent overtraining………………. ."

R1.5b) I strongly suggest authors update and modify the workflow diagram in this section (major comment).

Reply to R1.5b). Thank you for this valuable suggestion. We have revised the figure and added the relevant example of a CT image which explains the overall role of AI in Medical imaging. The updated text illustrating figures is pasted below.

R1.5c) I strongly suggest authors add the control parameters for AI applications in this section. (Major comment).

Reply to R1.5c). Thank you for your helpful suggestion. This work summarizes some representative published methods and shows example results. We have included the most relevant details and refer readers to published works for extensive details. Please also see the reply to 1.5a.

R1.6) Results and Discussions:

R1.6a) It has some figures, but technical description for the figures is not enough. The authors must describe as well for every figure (major comment).

Reply to R1.6a). Thank you for your valuable suggestion. The relevant text from the published work has been updated in the revised manuscript for all figures and pasted below:

“For Figure 2, As described in Jiang and Veeraraghavan, the model was trained using 20 unlabeled MRIs and an entirely different set of 30 patients with expert segmented CT scan images containing multiple organ segmentations. Testing was performed on set aside another group consisting of 10 patients who had undergone MRI exams. Both sequences were acquired on a 1.5 Tesla scanner. As shown, our approach produced a realistic synthesis of such images, indicating potential use in image harmonization. Synthesis realism was measured by computing the similarity of the features computed within the individual organs on synthesized images and those same organs in real images. Our method produced a low distance of 5.05 and 14.00 for T1w and T2w MRI. In comparison, this distance was 73.90 and 101.37 for T1w and T2w MRI using CycleGAN, which learns multiple one-to-one modality translations, and 73.39 and 77.49 for another state-of-the-art one-to-one modality translation method called StarGAN [20]."

“The results shown in Figure 3A, Figure 3B, and Figure 3C are produced by three different models that were trained using the CMEDL approach. Extensive details of the CMEDL method are in the prior published methods for CT lung tumor, [21] MRI lung tumor segmentation [17], and CBCT-based lung tumor segmentation [7]. Concisely, the CMEDL architecture makes use of two parallel segmentation subnetworks for a so-called trachker network (using MRI [in the case of Figure 3 A and 3B], and using CT [in the case of Figure 3 C]), and a student network (using CT [in the case of Figure 3A], CBCT [in the case of Figure 3B, and T2-weighted MRI [in the case of  Figure 3C. Any segmentation architecture can be used, as we showed using the popular Unet as well as a dense network called multiple resolution residual network (MRRN) [16]. The teacher network forces the student network to extract features that better signal the contrast between foreground and background by applying feature distillation losses that match the high-level features computed from corresponding synthesized teacher modality (e.g., MRI) and the student modality (e.g., CT) images.  The network itself is trained with unpaired images, images in which wherein corresponding sets of multiple modality scans are not required for training. To accomplish training with unpaired modalities, a cross-modality synthesis network created using a GAN is applied. The GAN consists of a generator created using a 3DUnet that computed dense pixel regression by using tanh activation, and a PatchGAN discriminator network to distinguish the synthesized from the real images was used in training. The details of the number of images used in training, training losses, as well as training epochs, etc. are in published methods [17]. The teacher network is initialized with example real images with and corresponding segmentations to learn to extract the appropriate set of relevant features. The same network is then jointly optimized with the student network to further refine the extracted features using synthesized images produced from the images input to the student network using the GAN-based image-to-image translation network. The teacher and student networks are jointly optimized during training to make use of multi-tasked optimization. The GAN network for synthesizing the cross-modality images is cooperatively optimized such that this network’s parameters are updated only in iterations when the segmentation network’s parameters are frozen and vice versa in order to ensure stable training convergence. Figure 3A shows the results of segmenting the tumor on CT scan images using a Unet network on a sample test case and optimized using with via the CMEDL approach using CTs (n = 377) and MRIs (n = 82) from external and internal institution datasets, respectively are shown in Figure 3A. Figure 3B shows tThe results of segmenting an ex-ternal institution CBCT scanimage using a Unet network optimized usingwith the CMEDL approach optimized with unpaired CBCTs (n = 216) and 82 MRIs from different sets of patients are shown in Figure 3B. Figure 3C shows a sample test set MRI segmenta-tion produced by training a Unet using the CMEDL approach. Separate models were constructed for the three results and optimized with different datasets. All networks were op-timized with the Adam optimizer, with an initial learning rate of 2e-4, batch size of 2, and early stopping used to prevent overfitting of the networks. As shown in the Figure 3, the algorithm generated segmentations (shown in red) closely approximates the expert delin-eation (shown in red) for the representative test cases.  WhileAlthough the aforementioned method focuses on the segmentation of the gross tumor volume (GTV), it is also important to consider the tumor margin needed to be included for effective treatment when using AI-defined tumors for treatment planning and delivery. For instance, in the context of thermal ablation, prior work by Singh et al. showed that incorporating blood perfusion information from dynamic contrast MRI using commercial software tools could be utilized to better define the margins of breast tumors for thermal ablation [22]. In the context of radiation therapy, the segmented GTV is often expanded to produce a clinical target volume (CTV) to incorporate the microscopic spread by using treatment planning software to generate automatic expansion with fixed criteria for different disease sites while accounting for limiting radiation exposure to the adjacent healthy tissues. However, this approach does not always account for microscopic disease and is resolved using a clinician’s manual delineation that leads to inter-rater variability [23]. Cardenas et al. [24] addressed this issue of clinical variability by using a stacked autoencoder deep network formulation to automatically learn the CTV definition for head and neck cancers while accounting for adjacent healthy tissues both for lymph nodes and high-risk CTV. A different prior work by Xie et al. [25] addressed the issue of lung cancer CTV definition by accounting for respiration and gross tumor volumeGTV contained within the CTV by constructing a customized loss function within a 3DUnet approach.”

R1.6b) this section needs to update and need more detail.

Reply to R1.6b). Thank you, the relevant text from the published work has been updated to clarify in the revised manuscript and is pasted below.

"Please see reply to R1.5a and  R1.6a."

R1.6c) The authors should add more detail and use more Graphs in these sections.

Reply to R1. 6c). Thank you for this helpful suggestion; the co-authors jointly decided to add relevant text from our published work to improve the clarity of the methods and results without adding additional graphs. "Please see reply to R1.5a and  R1.6a."

R1.7). Conclusion and recommendations:

R1.7a) Conclusion lack of novelty. Please rewrite your conclusion and add some highlight and novelty in it (major comment).

Reply to R1.7a). Thank you for this valuable comment. The conclusion has been updated and is pasted below:

" a few challenges and opportunities for AI application to cancer oncological imaging were summarized using novel methods for synthesizing different contrast modality images, auto-segmentation and image reconstruction with examples from lung CT and abdomen and head and neck MRI."

R1.7b) Conclusion is so short the authors should extend the material.

Reply to R1.7b. Thank you; the conclusion has been modified accordingly (see Reply 1.7a)

R1.8) Graphical abstract:

R1.8a) I think Graphical abstract can help researchers. I suggest authors add one Graphical abstract in the next version

Reply to 8a). Thank you;  we have added a graphical picture.

9) References:

9a) References should be updated (2021-2022)

Reply to 9a). Thank you; the relevant references have been updated accordingly in Table 1 and the main text of the revised manuscript.

References:

  1. Hosny, A.; Parmar, C.; Quackenbush, J.; Schwartz, L.H.; Aerts, H. Artificial intelligence in radiology. Nat Rev Cancer 2018, 18, 500-510, doi:10.1038/s41568-018-0016-5.
  2. Koh, D.-M.; Papanikolaou, N.; Bick, U.; Illing, R.; Kahn, C.E.; Kalpathi-Cramer, J.; Matos, C.; Martí-Bonmatí, L.; Miles, A.; Mun, S.K.; et al. Artificial intelligence and machine learning in cancer imaging. Communications Medicine 2022, 2, 133, doi:10.1038/s43856-022-00199-0.
  3. Razek, A.A.K.A.; Khaled, R.; Helmy, E.; Naglah, A.; AbdelKhalek, A.; El-Baz, A. Artificial intelligence and deep learning of head and neck cancer. Magnetic Resonance Imaging Clinics 2022, 30, 81-94.
  4. McCollough, C.; Leng, S. Use of artificial intelligence in computed tomography dose optimisation. Annals of the ICRP 2020, 49, 113-125.
  5. Lin, D.J.; Johnson, P.M.; Knoll, F.; Lui, Y.W. Artificial intelligence for MR image reconstruction: an overview for clinicians. J Magn Reson Imaging 2021, 53, 1015-1028.
  6. McLeavy, C.; Chunara, M.; Gravell, R.; Rauf, A.; Cushnie, A.; Talbot, C.S.; Hawkins, R. The future of CT: deep learning reconstruction. Clin Radiol 2021, 76, 407-415.
  7. Jiang, J.; Hu, Y.C.; Tyagi, N.; Zhang, P.; Rimner, A.; Deasy, J.O.; Veeraraghavan, H. Cross‐modality (CT‐MRI) prior augmented deep learning for robust lung tumor segmentation from small MR datasets. Medical physics 2019, 46, 4392-4404.
  8. Venkadesh, K.V.; Setio, A.A.; Schreuder, A.; Scholten, E.T.; Chung, K.; W. Wille, M.M.; Saghir, Z.; van Ginneken, B.; Prokop, M.; Jacobs, C. Deep learning for malignancy risk estimation of pulmonary nodules detected at low-dose screening CT. Radiology 2021, 300, 438-447.
  9. Bi, W.L.; Hosny, A.; Schabath, M.B.; Giger, M.L.; Birkbak, N.J.; Mehrtash, A.; Allison, T.; Arnaout, O.; Abbosh, C.; Dunn, I.F.; et al. Artificial intelligence in cancer imaging: Clinical challenges and applications. CA Cancer J Clin 2019, 69, 127-157, doi:10.3322/caac.21552.
  10. Huang, S.; Yang, J.; Fong, S.; Zhao, Q. Artificial intelligence in cancer diagnosis and prognosis: Opportunities and challenges. Cancer letters 2020, 471, 61-71.
  11. Diamant, A.; Chatterjee, A.; Vallières, M.; Shenouda, G.; Seuntjens, J. Deep learning in head & neck cancer outcome prediction. Scientific reports 2019, 9, 1-10.
  12. Liu, K.-L.; Wu, T.; Chen, P.-T.; Tsai, Y.M.; Roth, H.; Wu, M.-S.; Liao, W.-C.; Wang, W. Deep learning to distinguish pancreatic cancer tissue from noncancerous pancreatic tissue: a retrospective study with cross-racial external validation. The Lancet Digital Health 2020, 2, e303-e313.
  13. Jiang, J.; Elguindi, S.; Berry, S.L.; Onochie, I.; Cervino, L.; Deasy, J.O.; Veeraraghavan, H. Nested‐block self‐attention multiple resolution residual network for multi‐organ segmentation from CT. Medical Physics 2022.
  14. Jiang, J.; Hu, Y.-C.; Tyagi, N.; Rimner, A.; Lee, N.; Deasy, J.O.; Berry, S.; Veeraraghavan, H. PSIGAN: Joint probabilistic segmentation and image distribution matching for unpaired cross-modality adaptation-based MRI segmentation. Ieee T Med Imaging 2020, 39, 4071-4084.
  15. Jiang, J.; Hu, Y.-C.; Tyagi, N.; Zhang, P.; Rimner, A.; Mageras, G.S.; Deasy, J.O.; Veeraraghavan, H. Tumor-Aware, Adversarial Domain Adaptation from CT to MRI for Lung Cancer Segmentation. In Proceedings of the Medical Image Computing and Computer Assisted Intervention – MICCAI 2018, Cham, 2018//, 2018; pp. 777-785.
  16. Jiang, J.; Hu, Y.C.; Liu, C.J.; Halpenny, D.; Hellmann, M.D.; Deasy, J.O.; Mageras, G.; Veeraraghavan, H. Multiple Resolution Residually Connected Feature Streams for Automatic Lung Tumor Segmentation From CT Images. Ieee T Med Imaging 2019, 38, 134-144, doi:10.1109/TMI.2018.2857800.
  17. Jiang, J.; Rimner, A.; Deasy, J.O.; Veeraraghavan, H. Unpaired cross-modality educed distillation (CMEDL) for medical image segmentation. Ieee T Med Imaging 2021, 41, 1057-1068.
  18. Jiang, J.; Tyagi, N.; Tringale, K.; Crane, C.; Veeraraghavan, H. Self-supervised 3D anatomy segmentation using self-distilled masked image transformer (SMIT). arXiv preprint arXiv:2205.10342 2022.
  19. Jiang, J.; Veeraraghavan, H. Unified cross-modality feature disentangler for unsupervised multi-domain MRI abdomen organs segmentation. In Proceedings of the International Conference on Medical Image Computing and Computer-Assisted Intervention, 2020; pp. 347-358.
  20. Wu, P.-W.; Lin, Y.-J.; Chang, C.-H.; Chang, E.Y.; Liao, S.-W. Relgan: Multi-domain image-to-image translation via relative attributes. In Proceedings of the Proceedings of the IEEE/CVF international conference on computer vision, 2019; pp. 5914-5922.
  21. Gan, W.; Wang, H.; Gu, H.; Duan, Y.; Shao, Y.; Chen, H.; Feng, A.; Huang, Y.; Fu, X.; Ying, Y. Automatic segmentation of lung tumors on CT images based on a 2D & 3D hybrid convolutional neural network. The British Journal of Radiology 2021, 94, 20210038.
  22. Hoebel, K.V.; Patel, J.B.; Beers, A.L.; Chang, K.; Singh, P.; Brown, J.M.; Pinho, M.C.; Batchelor, T.T.; Gerstner, E.R.; Rosen, B.R.; et al. Radiomics Repeatability Pitfalls in a Scan-Rescan MRI Study of Glioblastoma. Radiol Artif Intell 2021, 3, e190199, doi:10.1148/ryai.2020190199.
  23. Cardenas, C.E.; Beadle, B.M.; Garden, A.S.; Skinner, H.D.; Yang, J.; Rhee, D.J.; McCarroll, R.E.; Netherton, T.J.; Gay, S.S.; Zhang, L. Generating high-quality lymph node clinical target volumes for head and neck cancer radiation therapy using a fully automated deep learning-based approach. International Journal of Radiation Oncology* Biology* Physics 2021, 109, 801-812.
  24. Cardenas, C.E.; McCarroll, R.E.; Court, L.E.; Elgohari, B.A.; Elhalawani, H.; Fuller, C.D.; Kamal, M.J.; Meheissen, M.A.; Mohamed, A.S.; Rao, A. Deep learning algorithm for auto-delineation of high-risk oropharyngeal clinical target volumes with built-in dice similarity coefficient parameter optimization function. International Journal of Radiation Oncology* Biology* Physics 2018, 101, 468-478.
  25. Xie, Y.; Kang, K.; Wang, Y.; Khandekar, M.J.; Willers, H.; Keane, F.K.; Bortfeld, T.R. Automated clinical target volume delineation using deep 3D neural networks in radiation therapy of Non-small Cell Lung Cancer. Phys Imaging Radiat Oncol 2021, 19, 131-137, doi:10.1016/j.phro.2021.08.003.

Reviewer 2 Report

Reviewer Comments for Manuscript ID: Cancers-2220189

The Manuscript titled “Artificial Intelligence in CT and MR Imaging for Oncological Applications” is well within the journal’s scope of Cancers. The authors summarized the role of Artificial Intelligence, including deep learning, machine learning in the context of CT and MRI imaging modalities. In entirety, the Manuscript is very well written, however, it needs some edits from a physics-point of view. Also, there are several suggestions for the authors to incorporate before recommending the work for publication. The authors should elaborate the Discussion section by including the advancements in medical imaging. The authors wherever throw some light on GLIOMAs (brain tumors), Abdomen, Lung, Head and Neck region. The authors should include breast cancer or prostate cancer also as the applications of AI are numerous. We also identified that authors didn’t discuss the Head and Neck region appropriately as summarized in abstract section. More concrete future research directions should be discussed. The authors must revise Future directions and Conclusions section.

Furthermore, per review guidelines, we find that the authors self-cited their work in large especially by the author Harini Veeraraghavan. Please consider removing or including the proper justification why the authors should include these respective studies. The following are identified:

[54] Citation belongs to Ramesh Paudyal, Amaresha Shridhar Konar, Amita Shukla-Dave

[19] Citation belongs to Usman Mahmood

[92] Citation belongs to Jaemin Shin

[20], [45], [60], [74] Citations belongs to Harini Veeraraghavan

There are further comments for the authors to improve the Manuscript.

(1) In reference to mentions from Lines 79;

AI tools represent a potential leap forward in oncologic imaging, including harness…

Line 91;

can be easily integrated into clinical oncologic imaging workflow, overcoming the time…….

This should be oncological.

Line 97;

the full spectrum of clinical, genomic, and histopathologic data in tumor classification [23]……

This should be histopathological.

(2) Page 4 of 16; Figure 1

Consider improving the quality of image in revised submission.

(3) Lines 192-199 and Figure 2

The authors mentions that their technique produced a realistic synthesis, however, they have not presented any quantitative evidence to support their claims. Compare and contrast with other published works.

(4) Section 4.2

Volumetric segmentation of tumor volumes and longitudinal tracking of tumor volume response

Lines 206-207

Reliable segmentation is also necessary to overcome the practical limitations of radiomics analysis methods, which require volumetric tumor segmentation.

The authors mentions that reliable volumetric tumor segmentation is necessary. There are software tools that mimics not only the three-dimensional tumour volume accurately from Magnetic Resonance Imaging or Compted Tomography but also precise ablation margins for surgeries and thermal interventions [https://doi.org/10.1016/j.cmpb.2020.105781]. Such tumor segmentation is based on the voxel information. Compare and contrast the development and applications of such extraction in relevant to customized Artificial Intelligence oriented deep learning/machine learning algorithms. Precisely, discuss the developments in reference to margins of tumors as they are critical for any clinical treatment.

Author Response

Reply to Reviewer #2 (R2) Comments: Manuscript ID: Cancers-2220189, Title: Artificial Intelligence in CT and MR Imaging for Oncological Applications

R2.1.1. The Manuscript titled "Artificial Intelligence in CT and MR Imaging for Oncological Applications" is well within the journal's scope of Cancers. The authors summarized the role of Artificial Intelligence, including deep learning, machine learning, in the context of CT and MRI imaging modalities. In entirety, the Manuscript is very well written; however, it needs some edits from a physics-point of view. Also, there are several suggestions for the authors to incorporate before recommending the work for publication.

Reply to R2.1.1. Thank you for thoroughly reading our invited review by Cancers and providing insightful comments to improve the quality. This manuscript is for a special issue of Cancers: Methods and Technologies Development. We have incorporated your valuable suggestions, and the changes are highlighted in the revised manuscript with track changes (reviewers' comments are in italics)

R2.1.2. The authors should elaborate the Discussion section by including the advancements in medical imaging. The authors wherever throw some light on GLIOMAs (brain tumors), Abdomen, Lung, Head and Neck region.

Reply to R2.1.2. Thank you for your helpful suggestion; the discussion section has been elaborated accordingly with appropriate references.

R2.1.3. The authors should include breast cancer or prostate cancer also as the applications of AI are numerous. We also identified that the authors didn't discuss the Head and Neck region appropriately, as summarized in the abstract section. More concrete future research directions should be discussed. The authors must revise the Future directions and Conclusions section.

Reply to R2.1.3. We agree with the reviewer's assessment; the relevant text and references have been added to the discussion section of the manuscript. The future research direction and conclusion sections are revised accordingly.

R2.1.4. Furthermore, per review guidelines, we find that the authors self-cited their work in large, especially by the author Harini Veeraraghavan. Please consider removing or including the proper justification why the authors should include these respective studies. The following are identified:

[54] Citation belongs to Ramesh Paudyal, Amaresha Shridhar Konar, Amita Shukla-Dave

[19] Citation belongs to Usman Mahmood

[92] Citation belongs to Jaemin Shin

[20], [45], [60], [74] Citations belongs to Harini Veeraraghavan

  • (19). Mahmood, U.; Shrestha, R.; Bates, D.D.B.; Mannelli, L.; Corrias, G.; Erdi, Y.E.; Kanan, C. Detecting Spurious Correlations With Sanity Tests for Artificial Intelligence Guided Radiology Systems. Frontiers in Digital Health 2021, 3, doi:10.3389/fdgth.2021.671015.
  • (20). Jiang, J.; Hu, Y.-C.; Tyagi, N.; Zhang, P.; Rimner, A.; Mageras, G.S.; Deasy, J.O.; Veeraraghavan, H. Tumor-Aware, Adversarial Domain Adaptation from CT to MRI for Lung Cancer Segmentation. In Proceedings of the Medical Image Computing and Computer Assisted Intervention – MICCAI 2018, Cham, 2018//, 2018; pp. 777-785.
  • (45). Jiang, J.; Hu, Y.C.; Tyagi, N.; Zhang, P.; Rimner, A.; Deasy, J.O.; Veeraraghavan, H. Cross‐modality (CT‐MRI) prior augmented deep learning for robust lung tumor segmentation from small MR datasets. Medical physics 2019, 46, 4392-4404.
  • (54). Do, RK; Reyngold, M.; Paudyal, R.; Oh, J.H.; Konar, AS; LoCastro, E.; Goodman, K.A.; Shukla-Dave, A. Diffusion-Weighted and Dynamic Contrast-Enhanced MRI Derived Imaging Metrics for Stereotactic Body Radiotherapy of Pancreatic Ductal Adenocarcinoma: Preliminary Findings. Tomography 2020, 6, 261-271, doi:10.18383/j.tom.2020.00015.
  • (60). Jiang, J.; Veeraraghavan, H. Unified cross-modality feature disentangler for unsupervised multi-domain MRI abdomen organs segmentation. In Proceedings of the International Conference on Medical Image Computing and Computer-Assisted Intervention, 2020; pp. 347-358.
  • (74). Jiang, J.; Rimner, A.; Deasy, J.O.; Veeraraghavan, H. Unpaired cross-modality educed distillation (CMEDL) for medical image segmentation. Ieee T Med Imaging 2021, 41, 1057-1068.
  • (92). Argentieri, E.; Zochowski, K.; Potter, H.; Shin, J.; Lebel, R.; Sneag, D. Performance of a Deep Learning-Based MR Reconstruction Algorithm for the Evaluation of Peripheral Nerves. In Proceedings of the Proceedings of the RSNA, 2019.

Reply to R2.1.4. Thank you for this comment; as suggested, reference [54] has been deleted in the revised manuscript. References [19] and [92] are relevant references for the application of AI in CT and MRI. Our novel published and unpublished results are described in section 3 of the review, and hence we need to cite these references [20], [45], [60], [74].

R2.2.1 There are further comments for the authors to improve the Manuscript.

Reply to R2.2.1. We appreciate Reviewer #2 suggestions to improve the quality of the manuscript.

R2.2.1a. In reference to mentions from Lines 79; AI tools represent a potential leap forward in oncologic imaging, including harness…

Reply to R2.2.1a. Thank you for pointing this out.; the text has been corrected in the revised manuscript.

R2.2.1b. Line 91; can be easily integrated into clinical oncologic imaging workflow, overcoming the time. This should be oncological.

Reply to R2.2.1b. Thank you, the text has been corrected in the revised manuscript.

R2.2c. Line 97; the full spectrum of clinical, genomic, and histopathologic data in tumor classification [23]. This should be histopathological.

Reply to R2.2.1c. Thank you, the text has been corrected in the revised manuscript.

R2.2.2. Page 4 of 16; Figure 1 Consider improving the quality of image in revised submission.

Reply to R2. 2. 2. Thank you for the helpful suggestion; Figure 1 is updated with a relevant CT example, reflecting the role of AI in Radiology clinical practice.

R2.2.(3) Lines 192-199 and Figure 2 The authors mentions that their technique produced a realistic synthesis; however, they have not presented any quantitative evidence to support their claims. Compare and contrast with other published works.

Reply to R2.2.3. Thank you for pointing this out. We have added relevant text in detail with appropriate references describing the networks in the revised manuscript for completeness. The added text in the revised manuscript is pasted below. The method detailed in the review is based on our published 2 journal articles [1,2] and two 8-page conference publications [3-8].

"Briefly, our method includes a domain invariant content encoder network composed of a sequence of convolutional layers and a single style coding network that extracts the latent style code for the different modalities. The style coding network is constructed using a variational autoencoder, which uses a latent Gaussian prior to span the styles of the various modalities and is constructed using 5 convolutional, pooling layers, followed by a global pooling and fully connected layer. The style code is transformed into a latent style scale by a latent scale layer that is then used to modulate the features computed by the decoder network to synthesize images corresponding to different modalities. This network is jointly optimized using adversarial losses using a patchGAN discriminator, content reconstruction loss, image translation losses, and latent code regression losses, as detailed in prior work [1]. In addition, a multi-tasked training strategy is used wherein which is a  two-dimensional 2D Unet architecture is employed to learn to generate multi-organ segmentation from the synthesized image sets. The networks are optimized using the Adam method, a batch size of 1, and learning rate of 2e-4, with early stopping used to prevent overtraining ……….."

R2.2(4) Section 4.2 Volumetric segmentation of tumor volumes and longitudinal tracking of tumor volume response

Reply to R2.2.4. Thank you, the relevant text has been added to the revised manuscript pasted below: "The results shown in Figure 3A, Figure 3B, and Figure 3C are produced by three different models that were trained using the CMEDL approach. Extensive details of the CMEDL method are in the prior published methods for CT lung tumor, [2] MRI lung tumor segmentation [3], and CBCT-based lung tumor segmentation [4]…………… ."

R2.2.5. Lines 206-207 Reliable segmentation is also necessary to overcome the practical limitations of radiomics analysis methods, which require volumetric tumor segmentation.

The authors mentions that reliable volumetric tumor segmentation is necessary. There are software tools that mimics not only the three-dimensional tumour volume accurately from Magnetic Resonance Imaging or Computed Tomography but also precise ablation margins for surgeries and thermal interventions [Pre-operative Assessment of Ablation Margins for Variable Blood Perfusion Metrics in a Magnetic Resonance Imaging Based Complex Breast Tumour Anatomy: Simulation Paradigms in Thermal Therapies, https://doi.org/10.1016/j.cmpb.2020.105781]. Such tumor segmentation is based on the voxel information. Compare and contrast the development and applications of such extraction in relevant to customized Artificial Intelligence oriented deep learning/machine learning algorithms. Precisely, discuss the developments in reference to margins of tumors as they are critical for any clinical treatment.

Reply to R2.2.5. Thank you for this important comment and for referring an appropriate reference which has been added to the revised manuscript. In addition, we have also cited references defining the clinical target volumes for radiation treatment.

We have illustrated our method with results for gross tumor volume definition and contrasted it with the improvements made by other methods by adding the following text in the revised manuscript pasted below:

 "Although the aforementioned method focuses on the segmentation of the gross tumor volume (GTV), it is also important to consider the tumor margin needed to be included for effective treatment when using AI-defined tumors for treatment planning and delivery. For instance, in the context of thermal ablation, prior work by Singh et al. [96] ………….. ."

References:

  1. Jiang, J.; Veeraraghavan, H. Unified cross-modality feature disentangler for unsupervised multi-domain MRI abdomen organs segmentation. In Proceedings of the International Conference on Medical Image Computing and Computer-Assisted Intervention, 2020; pp. 347-358.
  2. Gan, W.; Wang, H.; Gu, H.; Duan, Y.; Shao, Y.; Chen, H.; Feng, A.; Huang, Y.; Fu, X.; Ying, Y. Automatic segmentation of lung tumors on CT images based on a 2D & 3D hybrid convolutional neural network. The British Journal of Radiology 2021, 94, 20210038.
  3. Jiang, J.; Rimner, A.; Deasy, J.O.; Veeraraghavan, H. Unpaired cross-modality educed distillation (CMEDL) for medical image segmentation. Ieee T Med Imaging 2021, 41, 1057-1068.
  4. Jiang, J.; Hu, Y.C.; Tyagi, N.; Zhang, P.; Rimner, A.; Deasy, J.O.; Veeraraghavan, H. Cross‐modality (CT‐MRI) prior augmented deep learning for robust lung tumor segmentation from small MR datasets. Medical physics 2019, 46, 4392-4404.

Round 2

Reviewer 1 Report

The authors answered my concerns and challenges as well. This article in this version is acceptable.

Reviewer 2 Report

The manuscript is substantially improved after additions.